# From Metabolic Syndrome to Type 2 Diabetes in Youth

**DOI:** 10.3390/children10030516

**Published:** 2023-03-05

**Authors:** Dario Iafusco, Roberto Franceschi, Alice Maguolo, Salvatore Guercio Nuzio, Antonino Crinò, Maurizio Delvecchio, Lorenzo Iughetti, Claudio Maffeis, Valeria Calcaterra, Melania Manco

**Affiliations:** 1Department of Woman, Child and General and Specialistic Surgery, Regional Center of Pediatric Diabetes, University of Campania “L. Vanvitelli”, 80131 Naples, Italy; 2Pediatric Department, S. Chiara Hospital of Trento, APSS, 38122 Trento, Italy; 3Section of Pediatric Diabetes and Metabolism, Department of Surgery, Dentistry, Pediatrics, and Gynecology, University of Verona, 37129 Verona, Italy; 4Pediatric Unit, Department of Women’s and Children’s Health, S. Maria della Speranza Hospital of Battipaglia, 84091 Salerno, Italy; 5Center for Rare Diseases and Congenital Defects—Fondazione Policlinico Universitario “Agostino Gemelli” IRCCS, 00168 Rome, Italy; 6Metabolic Disorders and Diabetes Unit, “Giovanni XXIII” Children’s Hospital, A.O.U. Policlinico di Bari, 70124 Bari, Italy; 7Department of Medical and Surgical Sciences for Mothers, Children and Adults, Post Graduate School of Pediatrics, University of Modena and Reggio Emilia, 41121 Modena, Italy; 8Department of Internal Medicine, University of Pavia, 27100 Pavia, Italy; 9Pediatric Department, “V. Buzzi” Children’s Hospital, 20154 Milan, Italy; 10Preventive and Predictive Medicine Unit, Bambino Gesù Children’s Hospital, IRCCS, 00165 Rome, Italy

**Keywords:** adolescents, insulin resistance, insulin secretion, youth, metabolic syndrome, obesity, oxidative stress, type 2 diabetes

## Abstract

In the frame of metabolic syndrome, type 2 diabetes emerges along a continuum of the risk from the clustering of all its components, namely visceral obesity, high blood pressure and lipids, and impaired glucose homeostasis. Insulin resistance is the hallmark common to all the components and, in theory, is a reversible condition. Nevertheless, the load that this condition can exert on the β-cell function at the pubertal transition is such as to determine its rapid and irreversible deterioration leading to plain diabetes. The aim of this review is to highlight, in the context of metabolic syndrome, age-specific risk factors that lead to type 2 diabetes onset in youth; resume age specific screening and diagnostic criteria; and anticipate potential for treatment. Visceral obesity and altered lipid metabolism are robust grounds for the development of the disease. Genetic differences in susceptibility to hampered β-cell function in the setting of obesity and insulin resistance largely explain why some adolescents with obesity do develop diabetes at a young age and some others do not. Lifestyle intervention with a healthy diet and physical activity remains the pillar of the type 2 diabetes treatment in youth. As to the pharmacological management, metformin and insulin have failed to rescue β-cell function and to ensure long-lasting glycemic control in youth. A new era might start with the approval for use in pediatric age of drugs largely prescribed in adults, such as dipeptidyl peptidase-4 and sodium-dependent glucose transport inhibitors, and of new weight-lowering drugs in the pipeline such as single and multiple agonists of the glucagon-like peptide 1 receptor. The latter drugs can have tremendous impact on the natural history of the disease. By treating diabetes, they will reduce the burden of all the metabolic abnormalities belonging to the syndrome while causing a tremendous weight loss hitherto never seen before.

## 1. Introduction

The definition of metabolic syndrome (MetS) in youth remains unclear due to the absence of validated gold standard diagnostic criteria for the pediatric population [1]. Diagnostic criteria endorsed so far include the ones established by the National Cholesterol Education Program Adult Treatment Panel III (NCEP-ATP III) modified for age [2], the Weiss et al. criteria [3], the de Ferranti et al. criteria [4], and the Cruz and Goran criteria [5]. The latest criteria proposed by the International Diabetes Federation for youths aged 10–16 years include abdominal obesity (defined by increased waist circumference as ≥90th percentile or adult cut-off if lower) and two or more metabolic abnormalities (elevated systolic or diastolic blood pressure (SBP ≥ 130 mmHg or DBP ≥ 85 mmHg or treatment with anti-hypertensive medication); high fasting glucose (≥100 mg/dL) or overt type 2 diabetes (T2D); high triglyceride concentrations (≥150 mg/dL); reduced HDL cholesterol (<40 mg/dL)) [6].

The prevalence of MetS in children and adolescents varies from 0.2% to 38.9% [7], with a percentage of 3.3% in the general population, 11.9% in youth with overweight, and 29.2% in those with obesity [8]. The prevalence of MetS parallels often that of obesity, and, indeed, it is remarkably higher in high-income countries where obesity rates are higher. There is a wide variation in MetS prevalence based on age, gender, race/ethnicity, and criteria used for diagnosis [9].

Disturbances of the glucose homeostasis are central in the development and progression of the MetS, with insulin resistance (IR) being the common denominator for all the components of the syndrome. On the other hand, metabolic disorders clustering into the syndrome contribute to the worsening of the glucose homeostasis up to overt T2D, whose incidence in young ages is expected to rise in the next decades.

T2D in youth is diagnosed based on plasma glucose criteria, either a fasting plasma glucose (FPG) value ≥ 126 mg/dL (7.0 mmol/L) or a 2-h plasma glucose (2hPG) value ≥ 200 mg/dL (11.1 mmol/L) following a 75 g oral glucose tolerance test (OGTT). More recently, a value of glycated hemoglobin ≥ 6.5% (hbA1c, 48 mmol/mol) has been recognized as diagnostic criterion as well. T2D is also diagnosed in individuals with classic symptoms of hyperglycemia or hyperglycemic crisis, a random plasma glucose ≥ 200 mg/dL (11.1 mmol/L) [10].

T2D is becoming increasingly common and accounts for a significant proportion of youth-onset diabetes in at-risk populations. Children and adolescents with obesity, of certain ethnic and genetic backgrounds, and having a family history of T2D, are at the highest risk for T2D [11]. The SEARCH for diabetes in youth study reported a prevalence of 3916 individuals aged 10–19 years with T2D in the United States from 2002 to 2015. In Italy, a survey conducted in 2011 by the Diabetes Study Group of the Italian Society for Pediatric Endocrinology and Diabetology (ISPED) found a prevalence of T2D as high as 1% among new onset diabetes cases. Seventy percent of these 124 cases were asymptomatic at the onset [12]. The diagnosis was made based on incidental findings of fasting hyperglycemia (symptomatic in 20% of cases), hyperglycemia at 2 h following the OGTT in subjects with overweight/obesity, or on high fasting glucose [10].

On the other hand, in a large population of Italian children and adolescents with obesity of varying severity, the prevalence of T2D diagnosed as fasting, 2-h hyperglycemia, or high hbA1c was 0.4% [13].

In the present review, we will focus on T2D in the context of the MetS, with the aims of reviewing factors that inform the risk of developing T2D at an early age and at explaining why some individuals develop diabetes very early in life and some others do not; and discuss emerging treatments.

## 2. Risk Factors for T2D in Young Individuals with MetS

In adults with MetS, the transition from normal glucose homeostasis to diabetes takes decades, while in youths the evolution appears to be much more rapid [14]. Many factors are associated with the early development of T2D in young individuals with MetS, such as obesity, puberty, belonging to specific ethnic minorities, genetics, female gender, perinatal factors, nonalcoholic fatty liver disease (FLD), and polycystic ovary syndrome (PCOS).

### 2.1. Role of Visceral Adiposity, Inflammation, and Oxidative Stress

Obesity, especially visceral obesity, is one of the most important risk factors for the development of T2D, whose incidence increases linearly as a function of the obesity duration [15]. Up to 85% of adolescents with T2D live with overweight or obesity, and particularly with abdominal adiposity [15].

Young people with obesity have hyperinsulinemia and suffer peripheral resistance to the action of insulin (insulin resistance, IR) with an approximately 40% reduction in peripheral insulin sensitivity, meaning reduced glucose uptake in the muscle tissue and increased lipolysis in the adipose tissue [16,17].

The increased rate of inappropriate lipolysis in fasting and post-meal conditions results in an overflow of free fatty acids (FFAs) that accumulate ectopically in key metabolic organs (i.e., liver, muscle, heart, and pancreas). Adipose tissue is not just a fat storage organ but is the most expansive endocrine organ [15,18]. Adipose tissue can manifest low-grade inflammation and becomes fibrotic over time (“adiposopathy”). It releases multiple adipokines in the blood flow that can impair insulin sensitivity in peripheral organs [15].

These molecules are cytokines, such as interleukin-6 (IL-6) and tumor necrosis factor -α (TNF-α), and hormones such as leptin and resistin. High levels of inflammatory cytokines, in particular IL-6 and TNF-α, do not just cause reduced insulin sensitivity through the modulation of insulin signaling [17,19] but also have pro-apoptotic effects on the β-cell contributing to the progressive failure in insulin secretion that ends in plain diabetes [18]. In particular, levels of leptin are generally increased in subjects with obesity, because of the obesity-induced leptin resistance, which is thought to worsen IR. Conversely, the adiposopathy causes reduced secretion of adiponectin. This molecule mediates insulin action by improving insulin signaling and its levels are inversely related to the severity of IR. In young individuals living with obesity and T2D, levels of adiponectin are significantly reduced as compared with those with obesity but no diabetes and with normal weight controls [20]. Interestingly, while levels of leptin and adiponectin remain stable along the pubertal transition [21], puberty is generally associated with enhanced anti-oxidation capacity, particularly after an exercise bout in normal weight individuals [22]. The obesity condition impairs such anti-inflammatory capacity toward a more pro-oxidative pattern favoring oxidative stress. Pro-oxidation refers to mitochondrial and non-mitochondrial mechanisms, which generate reactive oxygen and nitrogen species. Anti-oxidation is the adaptive activation of enzymatic and non-enzymatic mechanisms that counterbalance pro-oxidation by activating the scavengers of pro-oxidants, and their products within cells and in extracellular body fluids. In pre-pubertal and pubertal boys with obesity, level of pro-inflammatory adipokines (leptin and IL-6) were correlated with those of pro-oxidative molecules such as thiobarbitouric acid reactive substances and protein carbonyls, while levels of adiponectin were associated to circulating anti-oxidative glutathione species and total antioxidant capacity (TAC). High-sensitivity C-reactive protein was increased and negatively associated with glutathione species. Such a pattern of association confirms the close association of obesity with low-grade inflammation and oxidative stress [23]. The pubertal transition may be a key time window for the onset of diabetes also because the burden that obesity imposes on the individual’s anti-inflammatory capacity toward a prevalent pro-inflammatory milieu. In that, some adipokines play a role favoring oxidative stress [24]. For instance, leptin promotes oxidation, increases phagocytic activity of macrophages, and induces synthesis of pro-inflammatory molecules. Indeed, pre-exercise levels of leptin predict reduced levels of post-exercise anti-oxidative glutathione species [24].

### 2.2. FFAs and Tissue Insulin Resistance

The release of FFAs from the adipose tissue is proportional to the subject’s fat mass. In individuals living with obesity, the adipose tissue is responsive to the lipolytic effect of catecholamines and relatively insensitive to the anti-lipolytic effect of insulin (“adipose tissue IR”). The insulin-dependent enzyme lipoprotein lipase, which limits the clearance of triglycerides and the leakage of FFAs from the adipocyte, has reduced activity in people with obesity, causing the hampered insulin-dependent esterification of FFAs in the adipose tissue [16]. 

In the muscle and the myocardium, FFA uptake is increased, leading to intra- and inter-fiber storage of lipids that, in turn, affect insulin signaling and result in reduced glucose uptake via down regulation of the glucose transporter 4 (GLUT4) receptor expression and in dysregulation of mitochondrial oxidation in muscle [25].

High plasma levels of FFAs also have a deleterious effect on the β-cell function (“lipotoxicity”) and cause, together with exposure to high levels of circulating glucose (“gluco-lipo-toxicity”), the progressive emergence of β-cell dysfunction in subjects genetically prone to diabetes [16,18].

The overflow of FFAs to the liver leads to an increased production and storage of triglyceride within the parenchyma up to an overt condition of FLD. The concomitant reduced synthesis of high-density lipoprotein (HDL) and very low-density lipoprotein (VLDL), and the reduced secretion of VLDL in the blood stream, favor the intrahepatic accumulation of fat. The overflow of FFAs also contributes to the increased rate of hepatic gluconeogenesis in the fasting condition (hepatic IR) [17].

### 2.3. Muscle–Adipose Tissue Cross-Talk

The relationship between muscle and adipose tissue was described by Raschke and Eckel as a two-edged sword [26]. They highlighted that some cytokines are released by the solo adipose tissue (i.e., adipokines) and some others by the solo muscle tissue (i.e., myokines) [26]. A number of cytokines are released by both. Known as adipo-myokines, the latter molecules serve importantly as mediators of exercise and inflammation and play a pivotal role to shape the risk of T2D onset from the very early ages of life [27]. The list of adipo-myokines includes IL-6, TNF-α, visfatin, myostatin, FSTL1, angiopoietin-like protein 4 (ANGPTL4), monocyte chemoattractant protein-1 (MCP-1), meteorin-like hormone (Metrnl), glypican-4 (GPC-4), and irisin. These molecules modulate different metabolic pathways as reported in a recent review by Graf et al. [27]. For instance, IL-6 promotes adipocyte lipolysis, FFA oxidation, and browning. Irisin stimulates adipocyte browning and lipolysis. Myostatin results in adipocyte lipolysis and mitochondrial lipid oxidation. “Browning” refers to the development of beige adipocytes that confer a healthier metabolic phenotype within the white adipose tissue characterized by increased expression of beige marker genes and increased presence of multilocular cells within the adipose tissue.

It is a matter of fact that adipokines modulate muscle metabolic pathways including glucose uptake through increased glucose transporter type 4 (GLUT4) translocation and expression, mitochondrial activity, and increased ability to take up and oxidize fat as a fuel [27], and, vice versa, myokines influence adipocyte metabolism. Importantly, exercise-induced adaptations include an altered profile of secreted myokines and adipokines that act in an endocrine manner to facilitate tissue-to-tissue communication and also the release of extracellular vesicles (EVs) by the adipose tissue [28]. EVs are membrane-derived nanoscale particles that convey proteins, lipids, nucleic acids, and metabolites as cargo material and take active part in dialogue between the adipose and the muscle tissues. Exercise can powerfully modulate this complex interplay of cytokines and EVs, with regard to their activities at rest and in post-meal conditions and how exercise might affect metabolic health in the long term. Contracting skeletal muscles release myokines that can alter the phenotype of the white adipose tissue toward the healthier phenotype of brown adipose tissue.

### 2.4. Role of Puberty

The Restoring Insulin SEcretion (RISE) study demonstrated with no doubt that the natural history of T2D in children is characterized by the more rapid decline of the β-cell function as compared with adults with obesity [29,30]. More severe IR in the former individuals might make the difference. Indeed, puberty is, in itself, a physiological state of IR that leads to the compensatory increase both in basal and stimulated insulin secretion [29]. The insulin-stimulated glucose uptake (i.e., the direct estimate of insulin sensitivity) as assessed by the hyperinsulinemic-euglycemic clamp is reduced by approximately 30% during Tanner stages 2–4 as compared to stage 1 and returns to the pre-pubertal value at the completion of the pubertal transition [29]. Reduced insulin sensitivity is physiologically compensated by increased insulin secretion in a hyperbolic relationship. The product of insulin sensitivity per secretion, also termed as disposition index, is the estimate of this relationship and is the most robust predictor of T2D incidence in youth [29]. In individuals with obesity, the demand on the β-cell to compensate for the increased IR at the pubertal transition may become challenging in those with severe obesity and/or in those who are genetically susceptible to diabetes, causing a reduced disposition index.

The additive effect on the IR condition of obesity and puberty can lead to the transition from a condition of compensatory hyperinsulinemia with normal carbohydrate tolerance to a state of inadequate insulin secretion due to dysfunctional β-cells manifesting with impaired glucose tolerance (IGT) or overt diabetes [29]. The median age of young patients who receive a diagnosis of T2D is about 13.5 years old, and this corresponds to the final stage of the pubertal transition. The onset of T2D in the pre-pubertal age should be considered a rarity.

Furthermore, the condition of obesity at the pubertal transition counters the beneficial increase of antioxidant capacity that characterizes early puberty in normal weight children as compared with pre-puberty [8]. Indeed, children living with obesity present with increased pro-oxidation and decreased anti-oxidation as compared with normal-weight peers at the pubertal transition [22].

### 2.5. Role of Ethnicity

Ethnicity is an important risk factor for the development of T2D in young ages. Most of the patients with T2D are from ethnic minorities. In the SEARCH for diabetes in youth study, the incidence of T2D in youths aged 10–19 years ranged from 7.7% per year in Asians/Pacific Islanders, to 6.5% per years in Hispanics, 6.0% per year in non-Hispanic blacks, and 3.7% per year in Native Americans [31]. In Caucasians, the prevalence of T2D is significantly lower, accounting for approximately 5.5% of adolescents with diabetes in the United States [32] and 1.5% in Europe [33].

African Americans and Native Americans have an insulin sensitivity reduced by 35–40% as compared with Caucasian adolescents of the same age, sex, and weight [32]. The Bogalusa Heart Study showed that, among ethnic groups, African Americans are the most insulin resistant ones [34].

### 2.6. Role of Genetic Susceptibility

Obesity is the most important risk factor for T2D. However, some adolescents with obesity do not develop T2D, and some others do develop the disease at a relatively lower body mass index (BMI) percentile than others, consistent with differences in the genetic susceptibility to this condition in the setting of obesity and IR [35]. T2D is a polygenic disease with heritability ranging from 30 to 70%. Single nucleotide polymorphisms (SNPs) that are significantly associated with T2D susceptibility are involved in the regulation of pivotal metabolic paths such as insulin sensitivity, insulin secretion, and adipogenesis [36]. They include, for instance, the Pro12Ala *peroxisome proliferator-activated receptor gamma* (*PPARG*) polymorphism [37], the Glu23Lys *KCNJ11* variant [38], and variants in *ABCC8*, *SLC2A2*, *HNF4A*, and *INS* that have been robustly associated with reduced insulin secretion [39]. The intronic variant rs7903146 (T allele) of *TCF7L2* is the SNP that is the strongest genetic determinant of T2D known so far, leading to an increased risk of T2D by 41% [40]. Genetic variants of *TCF7L2* influence pancreatic insulin secretion, and might influence T2D risk by modulating adipogenesis [41].

In African American youths, each copy of the T allele (rs7903146) in the TCF7L2 gene was associated with 1.97-fold (1.37, 2.82) increased odds for T2D [42].

In a multi-ethnic cohort of children with obesity, a genetic risk score (GRS) of five gene variants in the genes TCF7L2, IGF2BP2, CDKAL1, HHEX, and HNF1A, all known to modulate insulin secretion, was associated with lower insulin secretion and increased risk of progressing from normal glucose tolerance to IGT or diabetes [43]. In cross-sectional studies of normal-weight and overweight/obese children, GRSs associated with T2D and fasting insulin values, comprising 62 [44] or 53 SNPs [45], were associated with different glycemic traits, particularly with fasting glucose and estimates of beta cell function. In two cohorts of children and adolescents of Italian ancestry [46,47], the T2D risk genetic variants correlated to higher fasting glucose and insulin levels and to decreased β-cell function. A mediation analysis found obesity as an important factor modulating fasting glucose and insulin even in children and adolescents with normal weight [47].

### 2.7. Role of Perinatal Factors

The in utero life is paramount for shaping the offspring’s risk of T2D. Several factors can “program” such risk, mainly through epigenetic mechanisms. Children born to mothers with type 1 diabetes (T1D) or gestational diabetes have an increased risk of T2D onset [48]. Either low birth weight, which in most cases results from a delay of the intrauterine growth, or high birth weight are hallmarks of poor health during in utero life and both are associated with an increased risk of developing obesity, T2D, and all the features of the MetS [48]. Likely, either a redoubt or an excessive availability of nutrients during the in utero life cause metabolic and hormonal alterations (i.e., in the ratio between leptin and adiponectin levels at the placental site) that favor later-in-life IR, MetS, and also β-cell dysfunction [32,49]. Interestingly, the maternal intake of micro- and macronutrients during the pregnancy can also influence the offspring’s risk of obesity, IR, and impaired insulin secretion through epigenetic mechanisms. For instance, a low intake of polyunsaturated fats during pregnancy has been associated with differential methylation of genes involved in the onset of obesity and specifically of IR [50]. 

### 2.8. Role of Gender

IR is more severe and T2D rates are higher in girls from childhood to mid-puberty, whilst they are greater in boys during late puberty and adulthood. Accordingly, girls develop T2D at a higher BMI than boys [51].

Boys typically have greater central fat mass, particularly visceral adipose tissue, than girls, which is associated with a higher risk for T2D, through the mechanism of decreased insulin sensitivity and β-cell function [52]. Furthermore, reduced levels of testosterone in boys with obesity at the end of puberty might have a role since testosterone levels are robustly associated with the total antioxidant capacity [53]. On the contrary, fertile girls are thought to have higher antioxidant capacity than males. Studies in rodents have demonstrated that male rats have a higher degree of oxidative stress than female rats [54] and observations in humans demonstrate that there is a strong relationship between female estrogen levels and those of antioxidants [55].

### 2.9. Role of PCOS

PCOS is a common endocrine-gynecological disorder, affecting women of reproductive age, characterized by chronic anovulation, the ultrasound finding of polycystic ovaries, and clinical and biochemical hyperandrogenism. PCOS adolescents with obesity have a 50% reduction in insulin sensitivity in peripheral tissues, hepatic IR, and increased levels of circulating insulin owing to its reduced clearance [32]. A survey of adolescents with PCOS found a prevalence of IGT as high as 30% and of T2D as high as 4%. Adolescents with PCOS and IGT had a 40% reduction in first-phase insulin secretion compared with those with PCOS and normal glucose tolerance, confirming the pivotal role of hampered β-cell function for the progression toward overt diabetes [32].

### 2.10. Role of FLD

With the increase in the prevalence of obesity, FLD has become among the leading causes of chronic liver disease in the pediatric age. The disease affects the liver, but it also carries multiple extrahepatic manifestations, including T2D [56]. People with FLD have higher FPG, the hallmark of the hepatic IR, while the high amount of visceral fat that accumulates within the liver and the pancreas exacerbates the risk of progressing from high fasting glucose to overt diabetes. Children with biopsy-proven fatty liver have increased IR and hampered fasting insulin secretion, both correlating with the degree of fat deposition and the severity of hepatic inflammation within the context of the MetS [57]. In the cohort of 675 children with FLD (mean age of 12.6 years) from the National Institute of Diabetes and Digestive and Kidney Diseases NASH Clinical Research Network, the estimated prevalence of prediabetes was 23.4% (95% CI, 20.2–26.6%), and that of T2D was 6.5% (95% CI, 4.6–8.4%). Girls with FLD had 1.6 times greater odds (95% CI, 1.04–2.40) of having prediabetes and 5.0 times greater odds (95% CI, 2.49–9.98) of having diabetes than boys with FLD. The prevalence of steatohepatitis was higher in those with overt diabetes (43.2%) compared with prediabetes (34.2%) or normal glucose (22%) with an odds ratio of 3.1 as compared with those with normal glucose [58].

## 3. Screening for Prediabetes and T2D

While the screening for prediabetes and T2D in adults is cost-effective [59], the US Preventive Services Task Force recently concluded that, in asymptomatic children and adolescents, the evidence is insufficient to assess the balance of benefits and harms of screening for T2D, as the prevalence of T2D in children is low compared with adults [60]. Nevertheless, a risk-based screening for prediabetes or T2D is recommended in children and adolescents with overweight or obesity because of their close association [61]. In addition, T2D in youth is characterized by accelerated deterioration of insulin secretion and rapid development of complications [62]. Thus, it is crucial to screen young individuals with overweight and obesity for prediabetes and diabetes to ensure early diagnosis and targeted preventive and therapeutic interventions that could prevent or delay progression to T2D and development of its complications [63].

The guidelines of the American Diabetes Association (ADA) recommend considering testing youth over 10 years of age or after the onset of puberty, if it occurs earlier, with overweight (BMI > 85th percentile) or obesity (BMI > 95th percentile) and one or more risk factors for diabetes. Risk factors include (i) maternal history of diabetes or gestational diabetes mellitus during the child’s gestation; (ii) family history of T2D in first- or second-degree relative; (iii) high-risk race/ethnicity (i.e., Native American, African American, Latino, Asian American, Pacific Islanders); and (iv) signs of insulin resistance or associated conditions (acanthosis nigricans, hypertension, dyslipidemia, PCOS, or small-for-gestational-age birth weight). If tests are normal, it is recommended to repeat tests at a minimum of 3-year intervals or more frequently if BMI is increasing or the risk factor profile is deteriorating.

The assessment of FPG as a diagnostic tool for prediabetes and diabetes is feasible, suitable for all ages as it requires only a blood draw, and highly reproducible [64]. The disadvantage of the FPG measure is that it is influenced by stress and illness, both of which are of great relevance in pediatric age.

In contrast, HbA1c is an indirect measure of blood glucose and reflects average blood glucose levels over approximately 120 days [65]. Thus, HbA1c allows assessment of glycemic control over time, does not require fasting, and has greater pre-analytical stability and fewer daily perturbations due to stress, dietary changes, illness, or medication [66]. However, several physiological and pathological non-glycemic factors and conditions may alter HbA1c values (e.g., aging, sex, BMI, ethnicity, puberty, hemolysis, and renal function).

The OGTT consists of the collection of blood samples after the ingestion of a standard glucose load of 1.75 g/kg of body weight, up to a maximum dose of 75 g, after an overnight fast [67]. The 2hPG is used to assess glucose tolerance [67]. IGT is the hallmark of an impaired β-cell function in relation to reduced insulin sensitivity and is a high-risk condition for the development of T2D and cardiovascular disease (CVD) [66]. Thus, identifying patients with IGT is crucial, especially if FPG levels are within normal ranges [68]. The 1-h post-load glucose concentration (1hPG) during OGTT seems cost effective in identifying dysglycemia earlier than currently recommended biomarkers [69,70]. Specifically, a 1hPG level ≥ 155 mg/dl (8.6 mmol/L) may detect people with reduced β-cell function before the progression to IGT and diabetes and it is a more accurate predictor of the progression to diabetes than HbA1c or 2hPG levels in adults [69]. As regards pediatric age, a longitudinal study showed that, in youths with obesity, a value of 1hPG ≥ 155 mg/dl was associated with a greater reduction in β-cell function and, among the participants with normal value of 2hPG, with an increased risk of developing prediabetes over time [71].

Estimated prevalence of prediabetes and diabetes varies depending on the test used among FPG, 2hPG, and glycated hemoglobin. The use of one of these methods alone would fail to identify all individuals at risk of developing diabetes [31]. In the large population of children and adolescents from the “Bambino” study, there were very few cases of prediabetes meeting the two or even the three diagnostic criteria at the same time, and they were found in the teenage years, while no case was found in children before 10 years of age [13]. Therefore, use of all the three diagnostic criteria at the same time must be recommended for either screening or diagnostic purposes, particularly in teens. Routinely testing HbA1c at the point of care should be encouraged since it allows identifying young individuals at risk of T2D that otherwise would be missed. Indeed, children with HbA1c values greater than 5.7% showed a sevenfold (girls)/fourfold (boys) increased risk of developing T2D in adulthood [72].

However, so far, no algorithm has been validated for the risk-based screening of prediabetes and T2D in youth. In clinical practice, we continue to use diagnostic tests and threshold values that are derived from epidemiological evidence in adult populations, while it is still debated what test is the most suitable in clinical practice. Recently, Garonzi et al. have proposed a practical flow chart for the clinical use of screening tools in young individuals, taking into account pros and cons of each test [72]. They suggest starting to screen by testing FPG and HbA1c in the same fasting blood draw, as a first level examination. Measurement of FPG and HbA1c is convenient, requires only one blood draw, and can be arranged more easily, leading to a greater adherence to screening. Given the sensitivity and accuracy differences between HbA1c and FPG, simultaneous measurement of both increases the probability of identifying prediabetes and diabetes in children and adolescents [66,73]. In individuals with at least one altered result, they suggest performing an OGTT. The 2hPG testing could be also recommended for individuals with overweight or obesity from the age of 10 years or at the onset of puberty, if earlier, with at least one risk factor for T2D. Indeed, the use of FPG and HbA1c does not detect subjects with isolated postprandial hyperglycemia, who are at higher risk for early β-cell dysfunction and development of both T2D and CVD [74]. According to the consensus position statement of the ISPED and the Italian Society of Pediatrics, first level screening is suggested as early as 6 years of age in individuals with overweight and obesity [75]. This recommendation is based on the strength of a national prevalence of about 5% of prediabetes in children with obesity before 10 years of age. Considering these observations, it might be worth lowering the age of first-line screening and adjusting it according to the prevalence of glucose metabolism alterations in the reference population. When the diagnosis of T2D is being considered in children and adolescents with overweight/obesity, a panel of pancreatic autoantibodies should be performed to exclude T1D and, if clinically consistent, a genetic evaluation to exclude monogenic diabetes [66]. Differential diagnosis between T2D and Maturity Onset Diabetes of the Young (MODY) is not always easy, especially in the case of the co-presence of overweight and related metabolic alterations, which are becoming increasingly common due to the exponential prevalence increase in childhood obesity [76]. The lack of impact on blood glucose improvement of oral hypoglycemic therapy or weight loss, coupled with the finding of basal and during-OGTT insulin values in the normal range, or those inappropriately low for the degree of hyperglycemia, are useful clinical criteria that can guide to suspect a primary defect in β-cell function [77]. In case of atypical features or autosomal dominant inheritance, less common forms of diabetes should be considered in the differential diagnosis and, in this case, a custom panel including several genes associated to MODY and neonatal diabetes (i.e., HNF4A, GCK, HNF1A, PDX1, HNF1B, NEUROD1, KLF11, CEL, PAX4, INS, BLK, ABCC8, KCNJ11, APPL1, INSR, ZFP57, and PLAGL1) through Next Generation Sequencing analysis should be performed.

The prevalence of monogenic diabetes has been, and still is, underestimated as it is often undiagnosed or improperly classified as T2D, with negative consequences on prognosis and treatment. A recent study reported that more than 10% of T1D-related autoantibody-negative children with newly diagnosed diabetes had a genetic finding associated with monogenic diabetes [78]. The recognition of monogenic forms is crucial for the correct therapeutic approach and family counseling [77].

## 4. Diagnosis and Clinical Features at the T2D Onset

In Figure 1, we report the flow chart to diagnose T2D in youth.

According to the International Society of Pediatric and Adolescent Diabetes (ISPAD) [15], T2D occurs in young individuals because of obesity, when insulin production becomes inadequate with respect to the increased demand due to IR (relative insulin deficiency). As said, it is usually associated with other metabolic abnormalities characterized by increased IR, i.e., dyslipidemia, hypertension, PCOS, FLD, all of which make it possible, if persistent, to be framed within the MetS. Insulin secretion is variable and depends on the status and duration of the disease.

The main symptoms at the onset of T2D are the following: (a) up to 1/3 of cases onset with ketoacidosis: higher prevalence in at-risk ethnic groups and an average of 5–10%; (b) possible severe dehydration (hyperosmolarity); (c) M/F ranging from 1/4-6 North Americans and 1/1 Arabs. The increase in overweight and obesity in the general pediatric population creates troubles for the differential diagnosis with T1D. T1D is generally characterized by a lean phenotype, but in countries with a high prevalence of obesity, T1D also affects people with overweight and obesity, sometimes making the differential diagnosis cumbersome. In the case series of the Regional Center for Pediatric Diabetology “G.Stoppoloni,” which is located in Campania, the region with the highest prevalence of obesity in Italy and is the largest in the whole country for the number of referred patients in follow-up (about 1.100 up to 18 years of age) and of newly diagnosed per year (about 150), laboratory tests such as uric acid and triglyceride to HDL ratio (TG/HDL) were useful and practical means to discriminate T1D from T2D in patients with obesity at the disease onset. Figure 2 shows prevalence of T1D and T2D patients with high uric acid and TG/HDL ratio at the disease onset. In addition, a TG/HDL ratio > 2.2 was significantly correlated with hypertension, MetS, and increased waist circumference. These laboratory tests are easily available (in an admission profile) even before the results of autoimmunity markers, the positivity/negativity of which is, of course, diriment from a diagnostic point of view. Nevertheless, diagnostic accuracy of uric acid and the TG/HDL ratio must be investigated in an external population.

## 5. Treatment

The therapeutic approach to adolescents with T2D was defined by the Collaborative Consensus Guidelines among International Scientific Societies such as the American Academy of Pediatrics (AAP), Pediatric Endocrine Society (PES), Academy of Nutrition and Dietetics (AND), and American Academy of Family Physicians (AAFP) in 2012 and published in 2013 [79]. Thereafter, there were updates by the ISPAD in 2018 and more recently in 2022 [15,80] and, in the meantime, by the ADA in 2019 [81,82]. Updates take into account the rapid introduction to the market of novel therapeutic agents and molecules. The “clinical trials” that support the latest recommendations are represented by the Treatment Options for type 2 Diabetes in Adolescents and Youth (TODAY) Study [81,83] and from the more recent “Ellipse” (EvaLuation of Liraglutide in PediatricS with DiabEtes) phase 3 randomized clinical trial that defined the efficacy and safety of liraglutide in pediatric T2D [84].

The main goals of the treatment of children and adolescents with T2D are listed below: 1. Achieving and maintaining good glycemic control. 2. Improving endogenous insulin sensitivity and secretion. 3. Identifying and treating, if necessary, complications and comorbidities such as hypertension, dyslipidemia, and non-alcoholic hepatic steatosis. 4. Preventing vascular complications. 5. Avoiding unplanned pregnancies in adolescents with T2D because of the high risk of complications.

Therapeutic education is very important for achieving these goals. As emphasized by international guidelines, it is fundamental in the management of the child and adolescent with any diabetes, in particular with T2D. Ideally, the care of an adolescent with T2D should be managed by a multidisciplinary team at least consisting of a pediatric diabetologist, a nurse trained in diabetes management, a nutritionist/dietician, a psychologist, a social worker, and a sport medicine expert. First and foremost, all children with T2D should receive comprehensive education on self-management that includes teaching glycemic self-monitoring as long as needed and during periods of acute illness when symptoms of hyper- or hypoglycemia most often occur. Glucose monitoring or possibly even interstitial glucose sensor monitoring becomes imperative in patients on insulin therapy [85]. Just as for patients with T1D, HbA1c should be periodically monitored [79].

The focus of education for T2D is, initially and ongoing, on behavioral changes. Team members with expertise on dietary, exercise, and psychological needs of young people living with T2D should provide education [86] in a culturally sensitive, age-specific, and developmentally appropriate way to the patients and the entire family. The main scope is to empower the patient and the family, making them understand the principles of T2D treatment and the critical importance of the lifestyle changes. The diabetes education for families is based on a chronic care model. A comprehensive lifestyle program is recommended [84] since children living with T2D have been found to live a sedentary lifestyle that worsens their health complications. Lifestyle modifications can induce positive effects on health, reducing BMI, blood pressure, serum lipoproteins, and IR. Of note, strategies that impact positively on the teenager’s mental health are critical, as lifestyle modifications are not easily achieved and maintained over time, particularly when considering that young individuals living with diabetes may be under social, economic, and physical pressure, and are often affected by mental health disorders. In a short time, education becomes no longer efficient in ensuring glucose homeostasis control owing to difficulties in the adherence to the behavioral changes and to the faster progression of T2D in youths [87]. Lifestyle modifications are recommended at diagnosis [15,88] with the ultimate goal of 7–10% reduction in body weight in those who have completed linear growth or of achieving BMI < 85th percentile for age and sex for those who are still growing [89].

### 5.1. Nutritional Recommendations

Healthy eating patterns and habits must be encouraged [87,88,90], i.e., consumption of nutrient-dense high-quality foods and decreased consumption of calorie-dense, nutrient-poor foods as patterns and reduction in portion size, decreasing frequency of eating out, and replacing or eliminating high-calorie beverages [91]. Daily intake must be reduced by about 200 kcal per day while there is no recommendation in favor of any specific type of dietary regimen, since there is a lack of data regarding the effects on glycemic control in adolescents with T2D. Data on the short- or long-term efficacy of very low-calorie diets are lacking as well. Specific dietary intervention programs must be carried out under the supervision of an experienced nutritionist/dietitian to avoid macro- or micronutrient deficiencies. Gradual dietary changes in family habits must be recommended, and healthy parenting conducts related to diet and physical activity should be supported and encouraged. Dietary recommendations must be accustomed to the family’s cultural environment and financial constraints. As said, they must include [92,93] the elimination of sugar-sweetened soft drinks, including fruit juices; reduced intake of processed and prepackaged foods, and of refined, simple sugars and corn syrup; saturated and total fat intake; and conversely, increased intake of fruit and vegetables; of fiber-rich foods, such as whole grain products and legumes; of foods with low glycemic index; better portion control; and elimination of meals eaten away from home or while screen watching. Alike, family dietary behaviors that must be encouraged include the following: restricting the availability of high-fat, calorie-dense food and drink; understanding nutrition fact labels; emphasizing healthy parenting practices related to diet and activity but avoiding excessively restricted food intake; encouraging positive reinforcement of all goals achieved and avoiding blame for failure; promoting meals eaten on schedule, in one place, preferably as a family unit, avoiding screen activities or reading/studying while eating, and frequent snacking; maintaining food and activity logs as beneficial for raising awareness of food and activity issues and for monitoring progress. Patients are more likely to follow a diet that fits their preferences and habits and, therefore, dietary interventions must be personalized and finalized to goals that are both measurable and achievable. Frequent in-person visits with the dietitian (e.g., every 4 weeks) are recommended to assess progress and to keep both the patient and the family motivated [94].

### 5.2. Physical Activity

Physical activity plays a pivotal role in the management of young people living with T2D. It favors weight loss and more importantly enhances insulin sensitivity, hence improving blood glucose control [95,96]. Youth with T2D should be encouraged to engage in moderate (i.e., hiking, brisk walking, and skateboarding) to vigorous (i.e., soccer, basketball, ice or field hockey, jumping rope, and running) physical activity for at least 30–60 min at least 5 days per week [10] and at least 3 days of strength training per week. Resistance training (also called strength training) is recommended at least three times weekly and includes muscle-strengthening activities (i.e., weight lifting, push-ups, pull-ups, climbing ropes) and bone strengthening activities (i.e., skipping, running, and jumping rope). Both aerobic training and resistance training improve insulin sensitivity by enhancing muscle glucose uptake [96,97], and their combination has been found more effective for glucose control than either type of exercise alone [98]. Data are lacking on the effect of resistance training on glycemic control in youths with T2D, but evidence in those with overweight and obesity supports improvement in insulin sensitivity with resistance training in a way that is independent of changes in body composition [99,100].

Furthermore, regular physical activity leads to improvement in cardiovascular risk factors, well-being, and promotes weight loss [84].

Since many youths with T2D may be sedentary at the time of diagnosis, assessment of physical activity at baseline and a gradual increase to the target is recommended to avoid musculoskeletal injury and improve adherence.

It is also important to reduce prolonged sedentary behaviors and decrease non-academic screen time such as computers, televisions, and video games to less than 2 h daily [10].

The patient must engage in daily efforts to be more active physically, such as walking to school instead of taking the school bus or going by car, using stairs instead of elevators, and doing house and yard work. Physical activity must be regarded as a family event even though it has to be individualized for each patient. Encouraging a positive reinforcement of all achievements and avoidance of shaming is recommendable.

### 5.3. Drugs

Metformin and insulin were, in the recent past, the only drugs approved by the US Food and Drug Administration (FDA) for the treatment of T2D in children and adolescents, until FDA approval of liraglutide in June 2019 and of semaglutide in December 2022. Therefore, to date, these drugs are the pharmacological agents approved for treatment of T2D in children and adolescents in association with diet and exercise. Phentermine/Topiramate have been recently approved by the FDA but results are still controversial. Furthermore, the use of the drug has not been approved by the European Medicines Agency.

Metformin is the first-line therapy for most young patients with T2D in combination with non-pharmacological therapy. It is a biguanide that improves insulin response by increasing insulin-mediated glucose uptake in peripheral tissues and decreasing hepatic glucose production. Metformin has the additional benefit of causing modest weight loss. Results of the TODAY study indicate that most patients achieve a good initial response to the drug, but only about 50 percent of them maintain a persistent effective response over time [82].

Guidelines suggest that the initial regimen should depend on the degree of gluco-metabolic impairment. Metformin monotherapy is the choice for patients with HbA1c < 8.5%; <69 mmol/mol and no symptoms; combination therapy with metformin and basal insulin for patients with HbA1c ≥ 8.5%; ≥69 mmol/mol and symptoms of hyperglycemia (polyuria, polydipsia, nocturia, or weight loss), without ketoacidosis; and insulin alone for patients who have ketosis or ketoacidosis. Metformin should be added to the regimen only after ketoacidosis has been overcome and blood glucose values have returned to an acceptable range with insulin therapy.

In pediatric patients, metformin should be started with an oral dose of 500 mg administered once daily. The dose can be increased gradually with increases of 500 mg at one-week intervals until the maximum daily dose of 2000 mg is reached after four weeks. This dosage is generally administered as 1000 mg twice a day. Some pediatric diabetologists prefer to start with the higher dose of 1000 mg per day, but clinical experience suggests that slower titration may reduce gastrointestinal collateral effects (diarrhea, abdominal discomfort, abdominal pain). The drug should be taken with meals to reduce gastrointestinal symptoms. Metformin is contraindicated, because of the risk of lactic acidosis, in patients with hepatitis, cirrhosis, alcoholism, impaired renal function, or cardiopulmonary insufficiency, conditions that are extremely rare in pediatric age. Therefore, it is recommended to measure basal liver enzymes (alanine aminotransferase (ALT) and aspartate aminotransferase (AST)) and creatinine before starting metformin therapy. If liver enzyme levels are more than 2.5 times the upper limit of normal or if serum creatinine is markedly abnormal, it is recommended not to start metformin. In this situation, insulin therapy should be considered as the first choice.

Insulin therapy should be used soon at the onset as a first choice in patients who have ketoacidosis or severe hyper glycemia or in patients who have mixed characteristics of type 1 and type 2 diabetes (mixed forms or so-called “diabetes 1.5”). Insulin therapy is useful for these patients because they have usually inadequate insulin production (due to reduced β-cell function, which is also a consequence of the glucotoxic effect), as well as resistance to insulin itself. Guidelines recommend using insulin when blood glucose measured at random is ≥250 mg/dl or HbA1c is >9%, >75 mmol/mol [92].

Finally, some individuals with T2D require multi-injection insulin therapy that is the same as in T1D therapy. Because patients with T2D have IR, relatively high doses of insulin are required to restore glyco-metabolic control. A typical starting dose for insulin is 0.75–1.25 IU/kg/day up to 2 IU/kg/day. Glycemic control should be assessed by the self-monitoring of capillary blood glucose to adjust the insulin dose. Once ketosis has resolved and plasmatic glucose has returned to concentrations close to normal, metformin can be added. Some patients may then gradually switch from insulin therapy to metformin monotherapy. For those taking only basal insulin, the dose is reduced gradually within 2–6 weeks, provided that fasting blood glucose is maintained in the target interval (<130 mg/dl and ideally < 100 mg/dl). Patients who are unable to achieve this target range with metformin alone require combined therapy with basal insulin. Patients who initially require a regimen of multiple daily insulin injections with both fast- and slow-acting insulin analogs may gradually switch to a single daily dose of basal insulin. If glycemic goals are persistently maintained, the dose of insulin is reduced and eventually discontinued. Patients who do not achieve or make progress toward achieving glycemic control (ideally HbA1c < 7%; <53 mmol/mol fasting glucose < 130 mg/dl) require intensification of therapy [101].

Liraglutide, a GLP-1 (Glucagon-like peptide-1) analog, was approved by the FDA in 2019 [102] for use in pediatric patients with T2D, in therapeutic management in addition to diet and exercise to achieve good glycemic control, on the basis of a single clinical trial [92]. GLP-1 analogs (e.g., exenatide, liraglutide, semaglutide) are incretin mimetics that act to increase the glucose-dependent insulin secretion and help ensure adequate postprandial insulin response. These agents are administered by subcutaneous injection once daily. Liraglutide at the dose of 1.8 mg/day has the additional benefit of promoting modest weight loss, probably due to delayed gastric emptying and through central effects on appetite. The suggested starting dose of liraglutide is 0.6 mg subcutaneously once daily (via pen injector) for at least one week. The dose of liraglutide can be gradually increased by 0.6 mg per week up to the maximum dose of 1.8 mg/day to achieve fasting blood glucose goals of 130 mg/dl. The dose is titrated to 3 mg/day as a weight-loss-inducing medication. To minimize side effects, the dose should be increased slowly. If the patient is on insulin-associated treatment, the insulin dose should be reduced by 20 percent when starting liraglutide. After completing liraglutide dose titration, the insulin dose can be increased or reduced if necessary. Among liraglutide side effects, gastrointestinal symptoms (nausea or vomiting) are the most common (25–30% of patients) and usually occur during the first two months of therapy. Cases of severe acute pancreatitis and thyroid carcinoma have been reported as rare side effects. Therefore, liraglutide is not recommended in patients with a personal or family history of medullary thyroid carcinoma, multiple endocrine neoplasia type 2, or pancreatitis.

Similar to liraglutide, semaglutide is a GLP1 receptor agonist prescribed subcutaneously once-weekly at the maximal dose of 2.4 mg. It acts by decreasing appetite, thereby improving control of eating and reducing energy intake [103]. In the phase 3a randomized clinical trial, in association with lifestyle intervention, the medication led to a 16.7% reduction in BMI as compared with a 0.6% reduction in the placebo group after 68 weeks of treatment. Seventy-three percent of participants in the semaglutide group had a weight loss of 5% or more as compared with 18% of the participants in the placebo group at the end of the trial. A loss of body weight of at least 10% occurred in 62% of the participants in the semaglutide group and in 8% of the placebo group; a loss of at least 15% occurred in 53% and 5%, respectively; and a loss of at least 20% occurred in 37% and 3%, respectively. Additionally, 62% vs. 42% of participants in the two groups, respectively, reported gastrointestinal adverse effects, namely nausea, vomiting, and diarrhea. Serious adverse events were reported in the 11% of participants on semaglutide versus 9% on placebo [104].

Phentermine/Topiramate (PHN/TPM) (Qsymia^®^) is FDA-approved for adolescents ≥ 12 years with a BMI > 95th percentile for age and sex as an adjuvant therapy to lifestyle modifications [102,105]. Phentermine and topiramate seem to act to reduce appetite, to increase energy expenditure, and also to slow gastric emptying. Postulated mechanisms for appetite reduction include inhibition of norepinephrine reuptake, reduction of hypothalamic glutamate neurotransmission, and lowering neuropeptide Y levels [106,107,108]. PHN/TPM is a once-daily oral medication. The starting doses are 3.75 mg (phentermine) and 23 mg (topiramate) and are titrated every 14 days up to the maximum doses of 15 mg (phentermine) and 92 mg (topiramate). PHT/TPM should be used with caution in patients with a history of seizures, renal stones, or depression. Three serious adverse events have been reported in two studies randomized to top dose: suicidal ideation/depression and a bile duct stone [109].

A very unfortunate note at the end of this report has to be the fact that, at least at the present time, no other glucose-lowering drug has been approved in pediatric patients with T2D because of the lack of clinical trial data and the consequent safety concerns. In the TODAY study, rosiglitazone had been used as an alternative drug to metformin with good efficacy results. However, rosiglitazone was withdrawn from the market due to excessive side effects. Finally, we acknowledge that clinical trials are underway or about to start to evaluate the efficacy of various drugs in children and adolescents with T2D, such as dipeptidyl peptidase-4 (DPP-4) inhibitors and sodium-dependent glucose transport (SGLT2) inhibitors [110,111].

These molecules, however, serve as glucose-lowering agents while a great expectation is on poly-agonists that combine GLP-1 receptor agonism with glucose-dependent insulinotropic polypeptide (GIP) and glucagon receptor agonism. GLP1R agonists reduce energy intake by mechanisms that have not yet been fully unraveled in the hypothalamus and brain stem, slow the rate of gastric emptying, improve glucose-stimulated insulin secretion (GSIS), increase β-cell mass, and inhibit glucagon secretion. Similarly to GLP1 R agonists, GIPR agonists stimulate GSIS and increase β-cell mass, but, on the contrary, stimulate glucagon secretion. GIPR agonists also stimulate osteoblast activity while inhibiting bone resorption and increase postprandial adipose tissue blood flow. Glucagon receptor agonists exert various actions on liver paths. They increase glucose, fibroblast growth factor 21, and bile acid production; decrease de novo lipogenesis; and upregulate intrahepatic triglyceride lipolysis, ketogenesis, and fatty acid and amino acid oxidation [112]. They represent the most promising therapeutic approach to reduce body weight in a more effective way while improving insulin sensitivity, insulin secretion if still not yet fully compromised, and other pivotal metabolic paths. We are aware that 5% to 10% weight loss can be sufficient to improve the glucose profile and other metabolic abnormalities associated with obesity. These medications are able to cause ≥15% weight loss in a large proportion of patients and even ≥15% in a smaller group. Evidence in adult patients has proven reduced weight-lowering efficacy in patients with obesity and type 2 diabetes as compared with those with obesity and still preserved glucose homeostasis [113]. It is conceivable that adolescents with T2D would behave similarly to adults in terms of weight loss and, therefore, it might be very beneficial to prescribe these molecules very early in adolescents at risk of diabetes in order to prevent the onset of the disease. Furthermore, these molecules are effective in the long-term maintenance of healthy weight when in association with lifestyle intervention [113]. Chimera molecules in the pipeline for the treatment of obesity include molecules targeting glucagon receptors and inducing increased energy expenditure and reduced energy intake; the amylin beta cell hormone analog that is involved in both homeostatic and hedonic appetite regulation slows gastric emptying and thus suppresses postprandial glucagon responses to meals; and PYY analogs that also cause reduced energy intake, suppressing appetite [113].

## 6. Conclusions

Type 2 diabetes in youth appears to be firmly embedded within the metabolic abnormalities belonging to the syndrome owing to the pivotal role of IR. As compared with the disease in adults, this condition is much more quickly progressive, with the early failure of oral glucose-lowering agents to maintain the metabolic and glycemic homeostasis and early onset of severe CVD. Genetic susceptibility to the disease influences β-cell function in youths as well as in adults, but exposure during prenatal life and early life to different environmental factors can influence the individual’s epigenome, explaining the faster deterioration of insulin secretion. On the other hand, the pubertal transition with exaggerated insulin resistance pulls the trigger and unravels diabetes. In this frame, reducing and even preventing excessive weight gain is a promising strategy to reduce the burden of type 2 diabetes in young people.

## Figures and Tables

**Figure 1 children-10-00516-f001:**
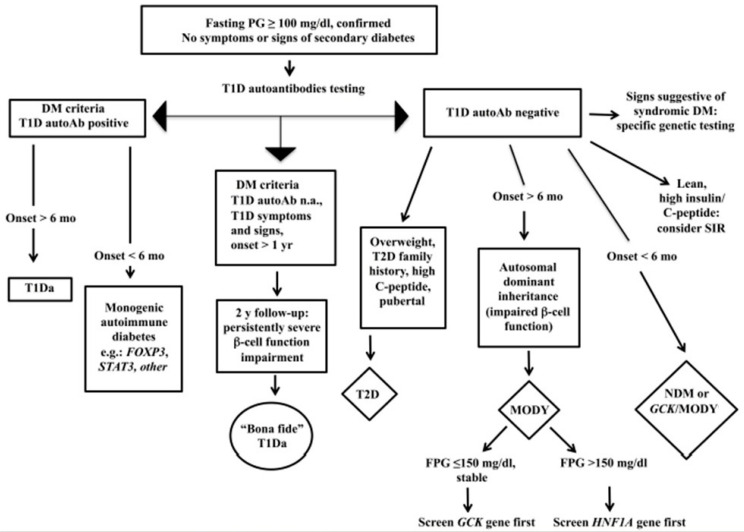
Flow chart for the diagnosis of pediatric diabetes.

**Figure 2 children-10-00516-f002:**
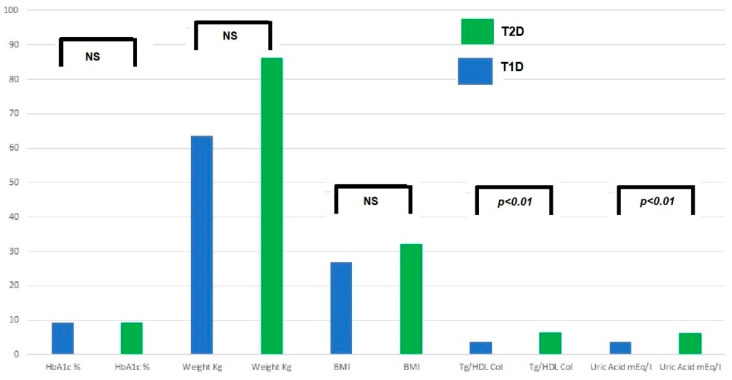
Prevalence of patients with high TG/HDL ratio and high uric acid.

## Data Availability

Not applicable.

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
