# Peer review of "From Metabolic Syndrome to Type 2 Diabetes in Youth"

_children, 2023, doi:10.3390/children10030516_

Round 1

Reviewer 1 Report

Overall Comment: In this narrative review Iafusco et al provide a comprehensive review of the literature on the development of T2D in children. I believe following revision of the following comments and detailed English editing of the manuscript, a useful review could be published.

Specific Comments

1. (pg 3 line 107-110): “The adipose tissue is not just the depot organ for fat storage, but is the largest endocrine organ that, once, becomes inflamed and fibrotic (“adiposopathy”) releases a large set of adipokines into the blood flow that, in turn, cause worsening of IR in peripheral organs” This phrase requires editing eg: “Adipose tissue is not just a fat storage organ, but can also be considered as the most expansive endocrine organ (REF:15). Adipose tissue can manifest low grade inflammation and become fibrotic overtime (“adiposopathy”) and release multiple adipokines in the blood flow that can affect IR in peripheral organs (REF:15).”

2. pg 3 line 125): Please add a reference

3. (pg 3 line 140): Please consider discussing the interplay of adipocytokines, inflammation and oxidation in children. Potentially you can refer to the study by Marseglia et al 2015 or  Paltoglou et al 2017.

4. (pg 3 line 141): Please expand the discussion on the interplay between adipose and muscle tissues.

5. (pg 5 line 225): Please consider discussing potential mechanisms regarding the effect of gender and sex hormones in mechanisms such as oxidation.

6. (pg 6 line 255): In this paragraph the authors provide a detailed analysis of the accelerator hypothesis and the role of risk factors in T1D development. In my view T1D is beyond the scope of the review and although the paragraph is well written the authors should strongly justify it’s inclusion or omit it.

7. (pg 7 line 311): “…type 1 diabetes risk..”

8. (pg 7 line 346): Please discuss the potential of HbA1c values in childhood as a risk factor.

9. (pg 8 line 395): Please when discussing the interesting flow-chart discuss the potential of genetic testing mode (panels) to identify MODY cases and the possibility of MODY patients sharing characteristics of T2D eg overweight.

10. (pg 9 line 427): Please also refer to the recently published 2022 ISPAD Guidelines.

11. (pg 13 line 630): Please refer to newer GLP-1 agonists that have recent FDA approval for adolescents (eg semaglutide) (Weghuber et al 2022)

12. (pg 13 line 630): Please discuss the potential of body weight lowering treatments to improve risk for developing T2D or metabolic syndrome.

Author Response

Reviewer #1

Overall Comment: In this narrative review Iafusco et al provide a comprehensive review of the literature on the development of T2D in children. I believe following revision of the following comments and detailed English editing of the manuscript, a useful review could be published.

Specific Comments

  1. (pg 3 line 107-110): “The adipose tissue is not just the depot organ for fat storage, but is the largest endocrine organ that, once, becomes inflamed and fibrotic (“adiposopathy”) releases a large set of adipokines into the blood flow that, in turn, cause worsening of IR in peripheral organs” This phrase requires editing eg: “Adipose tissue is not just a fat storage organ, but can also be considered as the most expansive endocrine organ (REF:15). Adipose tissue can manifest low grade inflammation and become fibrotic overtime (“adiposopathy”) and release multiple adipokines in the blood flow that can affect IR in peripheral organs (REF:15).”

Edited as suggested. Lines 113-116.

  1. pg 3 line 125): Please add a reference

Added as suggested.

  1. (pg 3 line 140): Please consider discussing the interplay of adipocytokines, inflammation and oxidation in children. Potentially you can refer to the study by Marseglia et al 2015 or  Paltoglou et al 2017.

Discussed as suggested quoting both studies. Lines 136-148. Furthermore we have introduced oxidative stress as key word and in the relative subheading on visceral adiposity (2.1 Visceral adiposity, inflammation and oxidative stress).

  1. (pg 3 line 141): Please expand the discussion on the interplay between adipose and muscle tissues.

Done. We have included the paragraph 2.3. Muscle-adipose tissue cross-talk (lines 173-203).

  1. (pg 5 line 225): Please consider discussing potential mechanisms regarding the effect of gender and sex hormones in mechanisms such as oxidation.

Done. See lines 227-231 & lines 329-300.

  1. (pg 6 line 255): In this paragraph the authors provide a detailed analysis of the accelerator hypothesis and the role of risk factors in T1D development. In my view T1D is beyond the scope of the review and although the paragraph is well written the authors should strongly justify it’s inclusion or omit it.

We agree and we have deleted the paragraph.

  1. (pg 7 line 311): “…type 1 diabetes risk..”

Corrected to T2D risk.

  1. (pg 7 line 346): Please discuss the potential of HbA1c values in childhood as a risk factor.

Done. See lines 384-387.

  1. (pg 8 line 395): Please when discussing the interesting flow-chart discuss the potential of genetic testing mode (panels) to identify MODY cases and the possibility of MODY patients sharing characteristics of T2D eg overweight.

Done.  See lines 414-432.

  1. (pg 9 line 427): Please also refer to the recently published 2022 ISPAD Guidelines.

Done. See line 472.

  1. (pg 13 line 630): Please refer to newer GLP-1 agonists that have recent FDA approval for adolescents (eg semaglutide) (Weghuber et al 2022).

Done. See lines 670-682.

  1. (pg 13 line 630): Please discuss the potential of body weight lowering treatments to improve risk for developing T2D or metabolic syndrome.

Done. See lines 703-731.

Reviewer 2 Report

The main contribution is the condensed information about the risk factors of Metabolic Syndrome and Type 2 Diabetes in children and adolescents. There is a lack of evidence in this population. From my point of view the abstract must content the aim of the review and it is not clearly presented. Also the conclusion can be improved . It would be important to broaden the conclusions by emphasizing risk factors and the transition from metabolic syndrome to type 2 diabetes in young people.

Author Response

The main contribution is the condensed information about the risk factors of Metabolic Syndrome and Type 2 Diabetes in children and adolescents. There is a lack of evidence in this population. From my point of view the abstract must content the aim of the review and it is not clearly presented. Also the conclusion can be improved. It would be important to broaden the conclusions by emphasizing risk factors and the transition from metabolic syndrome to type 2 diabetes in young people.

Done. See the abstract (lines 29-31 & 41-43) and the main text (intro: lines 91-94, and conclusions: lines 734-742).

Round 2

Reviewer 1 Report

After studying the report to the reviewers, I came to the conclusion that the authors have adequately answered all points raised and I would be happy to endorse the manuscript.